# Socio-demographic determinants of infectious disease-related health literacy and knowledge in Armenia: Results from a nationwide survey

Zhanna Sargsyan *, Zaruhi Grigoryan, Serine Sahakyan, Anya Agopian, Tsovinar Harutyunyan

Turpanjian College of Health Sciences, American University of Armenia, Yerevan, Armenia

* zhsargsyan@aua.am

**Data Availability Statement:** Data is publicly available here: DOI 10.17605/OSF.IO/TW2AX.

## Abstract

### Introduction

The success of health education programs heavily depends on the individuals' ability to comprehend information and apply it when making decisions regarding health. Low health literacy can lead to poor health in the context of both chronic and infectious diseases, as it influences a range of health-related behaviors. Little is known about infectious disease-related health literacy in Armenia and countries of similar socio-economic profiles. We aimed to quantify the levels of infectious disease-related health literacy and knowledge **a**mong the Armenian population and explore the underlying socio-economic factors.

### Materials and methods

We conducted a nationwide phone survey among 3483 adults selected via stratified two-stage probability proportional to size sampling in 2021. Descriptive statistics, simple and multivariable regression were used for the analysis.

### Results

The average age of respondents was 49.5 years. The majority were female (71.0%) and had at least 12 years of schooling (70.5%). The mean literacy score was 5.64 out of 7 (SD:1.06). The mean infectious disease knowledge score was 2.48 out of 4 (SD:1.19). Younger age, female gender, higher level of education, city residence, being employed, and having higher monthly expenditures were associated with higher mean scores of infectious disease-related health literacy and knowledge. In multivariable linear regression analysis, all the socio-demographic characteristics remained significant for both dependent variables.

### Conclusions

The study results revealed population groups with a higher infectious disease-related health literacy and knowledge. Interventions should focus on groups lagging behind to engage

**Funding:** This study received support from the United States Agency for International Development (USAID) to control COVID-19 and other infectious disease outbreaks (Cooperative Agreement # 72011120CA00003) United States Agency for International Development (USAID). This study is made possible by the generous support of the American People through the United States Agency for International Development (USAID). The contents of this paper are the sole responsibility of the American University of Armenia Fund and do not necessarily reflect the views of USAID or the United States Government. The funder had no role in study design, data collection and analysis, decision to publish, or preparation of the manuscript.

**Competing interests:** The authors have declared that no competing interests exist.

them in proper prevention practices to protect themselves and improve health status of Armenian population.

## Introduction

Infectious diseases are the leading causes of death worldwide and, despite the progress in infectious disease control, still jeopardize health and global security [1, 2]. Tuberculosis, influenza, measles, viral hepatitis, and other vaccine-preventable diseases remain responsible for a fair share of annual morbidity and mortality. The control and management of infectious diseases relies both on effective clinical care as well as on individual-level preventive practices [2]. The adoption of preventive practices, however, varies by the level of acceptance of the public health recommendations which, in turn, is mediated by the individuals' ability to comprehend the information. The latter is impacted by socio-demographic factors, educational background, and level of health literacy [2].

The health literacy refers to health-related knowledge and skills that are responsible for the acquisition and comprehension of health instructions and decision-making [2–6]. Various studies have concluded that low health literacy is linked to underutilization of preventive practices such as poor medication adherence, poor self-management, fewer healthy choices and risky health behaviors which eventually lead to poorer health outcomes [1, 3, 5, 7–10]. Specifically, the health literacy of infectious diseases has been found to be an important determinant of disease occurrence and health outcomes [11]. For instance, in the case of influenza, viral hepatitis, and COVID-19, low health literacy has been linked to being unvaccinated, having poor hand hygiene, and not maintaining distancing measures [2, 12, 13].

Interestingly, health literacy can also be negatively impacted by certain socio-demographic factors such as older age, low education level, and poor socio-economic status [3, 7]. For example, study that measured infectious disease-specific health literacy in Tibet, showed that low health literacy scores were associated with being female, older in age, having low education level, having other comorbid conditions, and being from an undeveloped area. The study found that, on the other hand, increasing literacy scores were associated with less frequent reports of infectious disease-related symptoms such as fever and diarrhea [7].

Furthermore, poor health literacy may put a heavy burden on healthcare systems as it can drain financial resources and unevenly exacerbate health outcomes among population groups that have low health literacy levels [5]. The promotion of health literacy can contribute to improved knowledge and behaviors which can thus enhance the resilience of communities to outbreaks of infectious diseases [10, 13].

Although Armenia has made great strides in the management of infectious diseases, they still impact morbidity and mortality in the country. In 2022, the incidences of Hepatitis B was 0.1 per 100,000 populations, hepatitis C was 0.4 per 100 000 populations, HIV was 18.0 per 100,000 population and tuberculosis was 9.8 per 100 000 populations. There were no cases of measles between 2020–2022, yet similar to the global rise in cases, the country experienced an outbreak in 2023 with 545 cases reported. The share of deaths due to infectious diseases (excluding COVID-19) in all-cause deaths was 0.6% in 2022 [14]. In order to ensure proper management and control of infectious diseases in the country, especially in the face of the re-emergence of many vaccine-preventable diseases, it is important to better understand infectious disease-related health literacy among the population. There are no studies to date that have measured infectious disease-related health literacy and knowledge among the Armenian

population. Therefore, we aimed to quantify the levels of infectious disease-related health literacy and knowledge to better understand the Armenian population's ability to comprehend and utilize health-related information for informed decision making. We explored the influence of socio-economic factors on infectious disease-related health literacy and knowledge, which might help with developing targeted health communication interventions for specific population groups.

## Materials and methods

### Study design and setting

The study team conducted a phone survey in the scope of larger study examining COVID-19 seroprevalence, which was commissioned by the USAID-funded "Support to control COVID-19 and other infectious disease outbreaks" project at the Turpanjian College of Health Sciences of the American University of Armenia (AUA/CHS) [15]. The primary objective of the study was to estimate the seroprevalence of antibodies against SARS-CoV-2 and assess the population's knowledge, attitude, and practices regarding infectious diseases transmission, manifestation, and treatment in general among adults in Armenia.

### Sampling and recruitment

The respondents were selected via stratified two-stage probability proportional to size sampling. During the first stage, one-third of the primary health care (PHC) facilities in each *marz* (region) and Yerevan were randomly selected based on the population size served by each facility. Simple random selection of the study participants was completed in stage two, with help of a countrywide electronic registry of patients (ARMED). Trained staff from selected PHC facilities received a list of potential participants from ARMED and, following standardized recruitment scripts via phone reached out to potential participants for recruitment into the study. The participants were recruited from different age groups proportionate to their size in the general population.

The initial sampling list provided to each facility by ARMED was six times larger than the needed sample. This was done to offset possible issues with sampling frame and nonresponse errors. The PHC staff continued recruitment until achieving the targeted sample size of 3483 participants. A more detailed description of participant recruitment, inclusion, and exclusion criteria have been explain in previous studies [16].

### Ethics statement

The Institutional Review Board of the American University of Armenia and World Health Organization Ethical Research Committee approved the study protocol (#AUA-2021-005; #WHO ERC- CERC.0013C). Data collection was conducted from May 27 –September 27, 2021. At the PHC facility, written informed consent was obtained to conduct blood sampling for antibody testing, and to acquire the contact information of the participants for the phone survey. All participants who provided written informed consent were contacted by interviewers within the first few days after the blood sampling to complete the phone survey.

### Data collection and study tool

Ten trained interviewers conducted interviews via electronic tablets using the Alchemer online tool (https://www.alchemer.com/). All participants who completed the phone survey were compensated for their time with a 1000 AMD (US$2.0) top-up of their phone cards.

The study protocol, including the processes of participant selection and enrollment and the administration of the survey, was pretested before fielding. A multi-domain survey questionnaire captured data on the participants' socio-demographic characteristics, infectious disease knowledge and infectious disease-related health literacy [7, 17, 18]. The infections disease-related health literacy questions were adapted from the WHO survey tool designed to guide behavioral insights studies related to COVID-19 to cover general infectious disease-related health literacy [17].

### Study variables

Socio-demographic characteristics of participants measured in the survey included place of residence (urban/rural), gender (male/female), age (continuous), education level (categorical), employment status (categorical), and average monthly expenditures (categorical).

Infectious disease knowledge was measured using four questions: 1)" Which *is the best way to be protected against measles?*"; 2) *"Antibiotics are effective treatment to cure flu."*; 3) *"Influenza is the same as common cold."; 4) "Influenza is caused by a virus."* The first question testing knowledge had multiple choice answer options with only one correct answer. The remaining three questions (numbers 2, 3, and 4) had answer options "Agree", "Disagree" and "Don't know". For these knowledge questions the correct answers were given 1 point and not correct answers and "Don't know" options 0 points. We added the total score of the four questions to come up with a summative infectious disease knowledge score with a range of 0 to 4.

Infectious disease- related health literacy was measured using nine questions: *"On a scale from 1 (very hard) to 7 (very easy), how easy or difficult would you say it is to:"*1) *"...find the information you need related to infectious diseases (for example, hepatitis, measles, influenza, COVID-19, HIV, etc.)?"*; 2) *"...understand information about what to do if you think you have an infectious disease (for example, measles, influenza, COVID-19, HIV, etc.)?"; 3) "...judge if the information about infectious diseases in the media is reliable?"; 4) "...understand restrictions and recommendations of authorities regarding infectious diseases?"; 5) "...follow the recommendations on how to protect yourself from infectious diseases?"; 6) "...understand recommendations about when to stay at home from work/school, and when not to?"; 7) "...follow recommendations about when to stay at home from work/school, and when not to?"; 8) "...understand recommendations about when to engage in social activities, and when not to?"; 9) "...follow recommendations about when to engage in social activities, and when not to?".* The scores for nine questions were added and the overall infectious disease mean score was calculated with a range of 1 to 7.

### Data management and analysis

The Statistical Package for Social Sciences (SPSS) version 21 (IBM Corp. Released 2012. IBM SPSS Statistics for Windows, Version 21.0. Armonk, NY: IBM Corp) was used for data analysis. The SPSS file was downloaded from the Alchemer platform [19]. The relationship between the infectious disease-related health literacy and infectious disease knowledge scores and the socio-demographic characteristics was examined by bivariate analyses (t-test). Cronbach alpha was used to assess the infectious disease-related health literacy scale for reliability. We performed two multivariable linear regression analyses to explore the adjusted relationship between the socio-demographic characteristics and the dependent variables: infectious disease-related health literacy and infectious disease knowledge scores.

### Results

The average age of respondents was 49.5 years. The majority lived in a city (88.5%), were female (71.0%) and had at least 12 years of schooling (70.5%). Over half (54.4%) of the respondents were employed (Table 1).

Table 1. Socio-demographic characteristics of the study participants.

| Variable | Total |
|---|---|
| | N (%) |
| | 3483 (100.0) |
| **Nationality** | |
| Armenian | 3442 (99.0) |
| Others | 36 (1.0) |
| **Place of Residence** | |
| City | 3075 (88.5) |
| Village | 400 (11.5) |
| **Gender** | |
| Male | 1011 (29.0) |
| Female | 2472 (71.0) |
| **Age (Mean, SD)** | 49.5 (14.8) |
| **Education** | |
| Less than 10 years to 10–12 years | 1022 (29.5) |
| Vocational (12–13 years) and University | 2382 (68.8) |
| Post graduate | 60 (1.7) |
| **Employment Status** | |
| Employed | 1882 (54.4) |
| Not Employed | 1577 (45.6) |
| **Monthly Expenditure** | |
| <100 000 AMD | 552 (19.8) |
| 101 000 AMD—400 000 AMD | 2117 (75.9) |
| >401 000 AMD | 119 (4.3) |

The response rate for each of the infectious disease-related literacy questions was high, ranging from 93.4% to 97.2%. Among the questions covering infectious disease-related health literacy, the question relating to respondents' ability to "judge if the information about infectious diseases in the media is reliable" had the lowest mean score (4.72, SD: 1.72) (Table 2). The highest mean literacy scores were for the questions concerning respondents' ability to "understand recommendations about when to stay at home from work/school, and when not to" (6.12, SD:1.32) and "understand recommendations about when to engage in social activities, and when not to" (6.12, SD:1.29). Overall mean literacy score, based on 3,040 respondents who answered all literacy questions, was 5.64 (SD:1.06). The literacy scale was found to have good internal consistency ($\alpha = 0.86$).

Respondents' knowledge of infectious diseases is presented in Table 3. Response rates were again high (above 97% for all), yet correct responses to the questions assessing infectious disease knowledge ranged widely from 49.4%, when asked about antibiotics as an effective treatment for the flu, to 77.9%, when asked whether influenza is caused by a virus. Overall mean infectious disease knowledge score, based on 3,372 respondents who answered all knowledge questions, was 2.48 (SD:1.19).

Bivariate analyses of the socio-demographic characteristics with both mean infectious disease-related health literacy and knowledge scores were assessed (Table 4). All variables, except nationality, were found to be significant and were included in the multivariate analyses to identify factors independently associated with each of the scores (Table 5). An increase in age ($\beta$: -0.005, 95% CI: -0.008 - -0.002) was found to be associated with a decrease in infectious disease-related literacy, while female sex ($\beta$: 0.288, 95% CI: 0.194–0.380), having a higher level

**Table 2. Infectious disease-related literacy.**

| Variable | Mean | SD | Response rate n (%) |
|---|---|---|---|
| **On a scale from 1 (very hard) to 7 (very easy), how easy or difficult would you say it is to:** | | | |
| Find the information you need related to infectious diseases (for example, hepatitis, measles, influenza, COVID-19, HIV, etc.)? | 5.25 | 1.77 | 3254 (93.4) |
| Understand information about what to do if you think you have an infectious disease (for example, measles, influenza, COVID-19, HIV, etc.)? | 5.43 | 1.61 | 3334 (95.7) |
| Judge if the information about infectious diseases in the media is reliable? | 4.72 | 1.72 | 3259 (93.6) |
| Understand restrictions and recommendations of authorities regarding infectious diseases? | 5.44 | 1.68 | 3342 (96.0) |
| Follow the recommendations on how to protect yourself from infectious diseases? | 5.87 | 1.43 | 3379 (97.0) |
| Understand recommendations about when to stay at home from work/school, and when not to? | 6.12 | 1.32 | 3381 (97.1) |
| Follow recommendations about when to stay at home from work/school, and when not to? | 5.94 | 1.42 | 3385 (97.2) |
| Understand recommendations about when to engage in social activities, and when not to? | 6.12 | 1.29 | 3382 (97.1) |
| Follow recommendations about when to engage in social activities, and when not to? | 5.95 | 1.39 | 3385 (97.2) |
| Infectious disease literacy summative score | 50.79 (max 63) | 9.50 | 3040 (87.3) |
| Infectious disease literacy mean score | 5.64 (max 7) | 1.06 | 3040 (87.3) |

(vocational or university) of education (β:0.314, 95% CI: 0.220–0.409), and a monthly expenditure of 101,000 to 400,000 AMD (β: 0.136, 95% CI: 0.026–0.246) were all found to be associated with increased infectious disease-related literacy in the adjusted analysis. A similar pattern was observed when examining factors associated with infectious disease knowledge. An increase in

**Table 3. Infectious disease knowledge.**

| Variable | Total n (%) |
|---|---|
| **Which is the best way to be protected against measles?** | |
| Vaccination | 2099 (61.57) |
| Other than vaccination | 217 (6.37) |
| Don't know | 1093 (32.06) |
| **Antibiotics are effective treatment to cure flu.** | |
| Agree | 1239 (36.6) |
| Disagree | 1672 (49.4) |
| Don't know | 475 (14.0) |
| **Influenza is the same as common cold.** | |
| Agree | 1166 (34.3) |
| Disagree | 1994 (58.7) |
| Don't know | 238 (7.0) |
| **Influenza is caused by a virus.** | |
| Agree | 2649 (77.9) |
| Disagree | 428 (12.6) |
| Don't know | 322 (9.5) |
| **Infectious disease knowledge score,** *mean (SD)* | 2.48 (1.19) |

**Table 4. Bivariate analysis of socio-demographic characteristics and infectious disease literacy and knowledge.**

| Variable | Infectious Disease Literacy Score Mean (SD) | p-value | Infectious Disease Knowledge Score Mean (SD) | p-value |
|---|---|---|---|---|
| **Nationality** | | | | |
| Armenian | 5.65 (1.06) | 0.507 | 2.47 (1.19) | 0.270 |
| Others | 5.52 (0.94) | | 2.69 (1.19) | |
| **Place of Residence** | | | | |
| City | 5.66 (1.05) | **0.011** | 2.50 (1.18) | **0.003** |
| Village | 5.51 (1.12) | | 2.31 (1.23) | |
| **Gender** | | | | |
| Male | 5.41 (1.16) | **<0.001** | 2.07 (1.20) | **<0.001** |
| Female | 5.73 (1.00) | | 2.64 (1.14) | |
| **Education** | | | | |
| Less than 10 years to 10–12 years | 5.36 (2.20) | **<0.001** | 2.03 (1.19) | **<0.001** |
| Vocational (12–13 years) and University | 5.75 (1.00) | | 2.65 (1.14) | |
| Post graduate | 5.72 (1.15) | | 2.91 (1.12) | |
| **Employment Status** | | | | |
| Employed | 5.74 (1.00) | **<0.001** | 2.61 (1.14) | **<0.001** |
| Not Employed | 5.52 (1.12) | | 2.32 (1.22) | |
| **Monthly Expenditure** | | | | |
| <100 000 AMD | 5.49 (1.15) | **<0.001** | 2.20 (1.22) | **<0.001** |
| 101 000 AMD—400 000 AMD | 5.71 (1.00) | | 2.56 (1.15) | |
| >401 000 AMD | 5.62 (1.00) | | 2.83 (1.31) | |

age was found to be associated with a decrease in infectious disease knowledge (β:-0.004, 95% CI: -0.007–0.001), while female sex (β:0.536, 95% CI: 0.440–0.632), having a higher level of education (vocational and university: β:0.505, 95% CI: 0.408–0.601; postgraduate: β:0.684, 95%

**Table 5. Multivariable regression analysis of socio-demographic characteristics and infectious disease knowledge and literacy.**

| Variable | Infectious disease literacy mean score | | | Infectious disease knowledge score | | |
|---|---|---|---|---|---|---|
| | **B-coeff** | **95% CI** | **p-value** | **B-coeff** | **95% CI** | **p-value** |
| **Age** | -0.005 | -0.008 0.002 | **<0.001** | -0.004 | -0.007–0.001 | **0.007** |
| **Gender** | | | | | | |
| Male | Ref | | | Ref | | |
| Female | 0.288 | 0.194 0.380 | **<0.001** | 0.536 | 0.440 0.632 | **<0.001** |
| **Place of residence** | | | | | | |
| City | Ref | | | Ref | | |
| Village | -0.054 | -0.177 0.699 | 0.395 | -.008 | -0.136 0.121 | 0.907 |
| **Education** | | | | | | |
| Incomplete/Secondary | Ref | | | Ref | | |
| Vocational and University | 0.314 | 0.220 0.409 | **<0.001** | 0.505 | 0.408 0.601 | **<0.001** |
| Postgraduate | 0.197 | -0.121 0.514 | 0.225 | 0.684 | 0.352 1.016 | **<0.001** |
| **Monthly Expenditure** | | | | | | |
| <100 000 AMD | Ref | | | Ref | | |
| 101 000 to 400 000 AMD | 0.136 | 0.026 0.246 | **0.015** | 0.221 | 0.110 0.332 | **<0.001** |
| >401 000 AMD | 0.056 | -0.166 0.277 | 0.621 | 0.515 | 0.281 0.748 | **<0.001** |
| **Employment Status** | | | | | | |
| Not Employed | Ref | | | Ref | | |
| Employed | 0.085 | -0.002 0.172 | **0.054** | 0.131 | 0.041 0.221 | **0.005** |

CI: 0.352–1.016), reporting higher monthly expenditures (101,000 to 400,000 AMD: β:0.221, 95% CI: 0.110–0.332; more than 400,000 AMD: β:0.515, 95% CI: 0.281–0.748), and being employed (β:0.131, 95% CI: 0.041–0.221) were found to be associated with increased infectious disease knowledge in the adjusted model.

## Discussion

Recognizing the importance of the impact of health literacy on preventive behaviors and population health, this study sought to assess infectious disease-related health literacy and knowledge in Armenia in an effort to gain insight into the current understanding of infectious diseases among the population. This study is the first to explore health literacy in Armenia, allowing for a unique examination into infectious disease-related health literacy and knowledge in relation to socio-demographic characteristics of the population.

The overall mean infectious disease-related literacy score was quite high (5.64 out of 7), which is encouraging. The lowest scores, on average, were obtained for the question asking respondents about their ability to "judge if the information about infectious diseases in the media is reliable". This could be reflective of a general mistrust in media among people living in Armenia that has also been observed around the world [20, 21]. The accessibility of diverse media platforms has provided the general public with access to varied and often conflicting information which may affect their trust in that information. The respondents understand recommendations concerning social settings in relation to infectious disease prevention, as the questions regarding when to stay home and when to engage in social activities resulted in the highest mean knowledge scores. These results are reassuring, especially as these were collected in the midst of the COVID-19 pandemic. It would be interesting to further examine whether this understanding impacted behaviors, and if so, in what way.

While infectious disease-related literacy scores were generally high, infectious disease knowledge scores were comparatively lower, with the overall mean score of 2.48 out of 4. These results taken together may be concerning, as it is likely respondents are able to access and believe they understand information on infectious diseases, yet their responses demonstrate that they may not be grasping the material correctly or that appropriate information is not being disseminated. Further examination into the information that is being accessed by the population as well as the way in which it is distributed could provide helpful information for future infectious disease awareness activities.

A number of socio-demographic factors were found to be associated with both infectious disease-related literacy and knowledge, including age, gender, education, and monthly expenditure. Similar to previous studies, as respondents' age increased, their literacy and knowledge scores decreased [22, 23]. This may be due to a lower education levels in older adults, although a majority (70.5%) of our sample had at least a university or vocational degree. It may also be that older adults are less connected to various forms of information, including the internet, and thus are not exposed to current health-related information. Information campaigns regarding infectious diseases should be targeted toward older Armenians, as they are among those most at risk. Moreover, women were found to have higher literacy and knowledge scores. Studies have mixed findings on the association of gender with health literacy [3, 4]. Notably, the results observed in this study may be reflective of the large proportion of respondents who are female (71%). Further examination of gender differences may provide helpful insight to improve literacy and knowledge among the male population. The results also indicated that respondents with higher education and monthly expenditures had higher literacy and knowledge scores, along with higher knowledge scores among those who were employed. These are all consistent with the previous literature [7, 23, 24], and thus highlight the need to

reach out to those of lower educational and economic levels to boost overall infectious disease knowledge and literacy in the Armenian population.

It is important to note that the data used in this study were collected as part of a survey examining COVID-19 knowledge, attitudes and practices, which could impact the results seen. Due to the pandemic, people globally became much more aware of infectious disease transmission and prevention, either by actively following the evolution of the pandemic or by the mere fact that it occupied so much of the conversation in all spheres of life. In Armenia, mass public information campaigns aimed at raising awareness about the transmission and prevention of COVID-19 were implemented, leading to a potential for increase in infectious disease-related literacy and knowledge.

The study does have a few limitations. As mentioned previously, the data used were collected in the context of a study examining knowledge, attitudes and practices towards COVID-19, which may have, in fact, positively impacted the results. Also, those more concerned about their health and/or more aware of infectious diseases might agree to participate in the blood sampling for COVID-19 antibody testing introducing some degree of self-selection bias. Furthermore, since the survey was not focused on health literacy, the study was constrained by the information that was collected, limiting the availability and thus ability to examine other potential socio-demographic factors associated with literacy and knowledge of infectious diseases. And finally, the scale used to assess infectious disease-related health literacy was not previously validated. Future studies may consider using validated scales and designing comprehensive questionnaires specifically focusing on the infectious disease-related health literacy and knowledge. In order to ensure the results from the scale provided valid information, internal consistency of the scale was assessed and found to be acceptable, thus allowing for a sense of assurance in the results of the analyses using the literacy score.

## Policy implications

Infectious disease-related literacy and knowledge are key for their prevention and transmission. By assessing these among the Armenian population, the study found that while literacy was generally high, knowledge could bear improvement. In order to increase both among the population, interventions and other activities should focus on those who are older and from lower socioeconomic backgrounds. Policymakers should focus on developing and implementing targeted educational campaigns to bridge the gap between understanding and knowledge, as well as addressing areas of misconception identified in this study. Suggested improvement mechanisms include prioritization of the concept of health literacy at the national level along with the involvement of various major stakeholders such as the general public, community-based organizations, healthcare facilities, media, academic community and business community [5]. Higher infectious disease-related literacy and knowledge will help ensure communities are empowered to engage in proper prevention practices to protect themselves and the Armenian population at large. The findings of this study also highlight the global relevance of health literacy in infectious disease prevention and control. International organizations and policymakers should advocate for the prioritization of health literacy on the global health agenda, emphasizing its crucial role in empowering communities to adopt proper prevention practices and respond effectively to future infectious disease outbreaks.

## Author Contributions

**Conceptualization:** Serine Sahakyan, Tsovinar Harutyunyan.

**Data curation:** Zhanna Sargsyan, Serine Sahakyan.

**Formal analysis:** Zhanna Sargsyan, Zaruhi Grigoryan, Anya Agopian.

**Funding acquisition:** Serine Sahakyan, Tsovinar Harutyunyan.

**Investigation:** Serine Sahakyan, Tsovinar Harutyunyan.

**Methodology:** Serine Sahakyan, Tsovinar Harutyunyan.

**Project administration:** Zhanna Sargsyan, Serine Sahakyan, Tsovinar Harutyunyan.

**Supervision:** Serine Sahakyan, Tsovinar Harutyunyan.

**Visualization:** Zhanna Sargsyan.

**Writing – original draft:** Zhanna Sargsyan.

**Writing – review & editing:** Zhanna Sargsyan, Zaruhi Grigoryan, Serine Sahakyan, Anya Agopian, Tsovinar Harutyunyan.

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
