## [Decision Letter · Decision Letter 0]

8 Mar 2024

PONE-D-23-40060Socio-demographic determinants of infectious disease-related health literacy and knowledge in ArmeniaPLOS ONE

Dear Dr. Sargsyan,

Thank you for submitting your manuscript to PLOS ONE. After careful consideration, we feel that it has merit but does not fully meet PLOS ONE’s publication criteria as it currently stands. Therefore, we invite you to submit a revised version of the manuscript that addresses the points raised during the review process.

**ACADEMIC EDITOR: Please address the reviewers' feedback to improve your manuscript.**==============================

We look forward to receiving your revised manuscript.

Kind regards,

Laura Brunelli, MD, PhD

Academic Editor

PLOS ONE

 [Support to control covid-19 and other infectious disease outbreaks. Cooperative Agreement # 72011120CA00003. United States Agency for International Development (USAID)].  

Reviewers' comments:

Reviewer's Responses to Questions

**Comments to the Author**

1. Is the manuscript technically sound, and do the data support the conclusions?

Reviewer #1: Yes

Reviewer #2: Yes

2. Has the statistical analysis been performed appropriately and rigorously? 

Reviewer #1: Yes

Reviewer #2: Yes

3. Have the authors made all data underlying the findings in their manuscript fully available?

Reviewer #1: Yes

Reviewer #2: No

4. Is the manuscript presented in an intelligible fashion and written in standard English?

Reviewer #1: Yes

Reviewer #2: Yes

5. Review Comments to the Author

Reviewer #1: Dear Authors,

This article deals with a very interesting topic. The relevance of this article is undeniable. Additionally, the article is well structured, the methods of analysis are relevant, and the tables are informative. However, before its publication, some changes need to be made.

You will find my comments below.

Sincerely yours

1. The introduction needs polishing. The authors should add statistics on infectious diseases in Armenia.

2. The authors mention that ‘The study team conducted a phone survey in the scope of existing larger study of COVID-19…’. What is the objectives of this phone survey ?

3. The authors said that ‘The infectious disease knowledge was measured using four questions.’ I suggest them to add the four questions.

4. ‘The infectious disease-specific health literacy was measured using nine questions’. Same comment as above.

5. In the discussion section do not use ‘It seems’. Use other words. ‘It seems’ is not a scientific word in these sentences.

6. I suggest to authors to refine the discussion. For example, to explain the effect of age, you need to take into account the differences in educational levels between younger and older people. You need to mobilize data and other studies on Armenia's socio-cultural context.

7. Finally, The authors need to add limitations regarding the type of survey (phone survey). In addition, I suggest that the authors add a section ‘Policy implication of findings’

Reviewer #2: Employing a comprehensive nationwide phone survey, the study endeavours to measure the extent of health literacy and knowledge concerning infectious diseases across Armenia, while also examining the impact of socio-economic factors on these variables.

The article is generally well-written, and it includes a thorough review of the literature, presents a sound methodology, and discusses the findings in a comprehensive manner.

Some minor revisions can be addressed to enhance the credibility and interpretability of the research findings; details are reported below.

- I believe that an evidence-based definition of “working from home” and a consistent use of the related terminology for this exposure.

- The study design should be understood from the title and the abstract; it should bel also mentioned in the methods section.

- Even though the language adopted in the article is correct, I suggest refining the style to achieve better clarity and precision. Be careful in ensuring that the terminology used is the most accurate and specific to avoid ambiguity.

- The methodology is well-structured, employing a nationwide phone survey to collect data. However, the study could benefit from a more detailed description of the survey questions and the rationale behind the selection of these specific questions. Clarification on how the questions were tailored to assess health literacy and knowledge specifically related to infectious diseases would strengthen the paper.

- It's mentioned that the scale used to assess infectious disease-related health literacy was not previously validated, which could affect the reliability of the results. If possible, use a validate scale or, in future studies, consider validating the scale used in this paper.

- While the paper acknowledges several limitations, including the potential bias introduced by phone surveys and the lack of validation for the health literacy scale, a more thorough exploration of these limitations and their implications for the study's findings would be beneficial. Additionally, discussing potential strategies for overcoming these limitations in future research could provide a more comprehensive picture.

- A detailed participant flowchart or description of participant recruitment, inclusion, and exclusion criteria would enhance transparency in reporting.

- The discussion effectively highlights the importance of the study findings in the context of Armenia and beyond. However, the paper would benefit from a broader consideration of the implications for public health policy and education strategies. Specifically, recommendations for how policymakers and educators can use these findings to improve health literacy and preparedness for future infectious disease outbreaks would be useful.

- While the paper acknowledges several limitations, including the potential bias introduced by phone surveys and the lack of validation for the health literacy scale, a more thorough exploration of these limitations and their implications for the study's findings would be beneficial. Additionally, discussing potential strategies for overcoming these limitations in future research could provide a more comprehensive picture.

6. PLOS authors have the option to publish the peer review history of their article (what does this mean?). If published, this will include your full peer review and any attached files.

Reviewer #1: No

Reviewer #2: No

---

## [Author Response · Author response to Decision Letter 0]

7 Jun 2024

Dear Editorial Team, 

Thank you for the review and the constructive feedback on our manuscript entitled “Socio-demographic determinants of infectious disease-related health literacy and knowledge in Armenia”.

On behalf of the authors’ team, I would like to upload the revised version of the manuscript which addresses the reviewer’s comment. Please find our response below. 

Thank you, we followed the guidelines while preparing the revisions.

 [Support to control covid-19 and other infectious disease outbreaks. Cooperative Agreement # 72011120CA00003. United States Agency for International Development (USAID)]. 

Support to control COVID-19 and other infectious disease outbreaks. Cooperative Agreement # 72011120CA00003. United States Agency for International Development (USAID).

“This study is made possible by the generous support of the American People through the United States Agency for International Development (USAID). The contents of this paper are the sole responsibility of the American University of Armenia Fund and do not necessarily reflect the views of USAID or the United States Government.”

The participants of this study did not agree to sharing their data publicly in a non-aggregated format; thus, in accordance with the study’s IRB approval and participant consent, the data is unable to be made available for public use. 

 

Reviewer #1: 

Dear Authors,

This article deals with a very interesting topic. The relevance of this article is undeniable. Additionally, the article is well structured, the methods of analysis are relevant, and the tables are informative. However, before its publication, some changes need to be made.

You will find my comments below.

Sincerely yours

Thank you!

1. The introduction needs polishing. The authors should add statistics on infectious diseases in Armenia.

We polished the introduction section and added statistics on most common infectious diseases in Armenia. (lines 38-103)

2. The authors mention that ‘The study team conducted a phone survey in the scope of existing larger study of COVID-19…’. What is the objectives of this phone survey ?

We added the primary objective of the larger study. 

“The primary objective of the study was to estimate the seroprevalence of antibodies against SARS-CoV-2 and assess the population’s knowledge, attitude, and practices regarding infectious diseases transmission, manifestation, and treatment in general among adults in Armenia.” (lines 107-110)

3. The authors said that ‘The infectious disease knowledge was measured using four questions.’ I suggest them to add the four questions.

We added the four questions in the methods section. (lines 151-155)

4. ‘The infectious disease-specific health literacy was measured using nine questions’. Same comment as above.

We added the nine questions. (lines 160-175)

5. In the discussion section do not use ‘It seems’. Use other words. ‘It seems’ is not a scientific word in these sentences.

Thank you for the comment. We replaced the word “it seems” with “it is likely” (line 264) and removed word “seem” (line 256).

6. I suggest to authors to refine the discussion. For example, to explain the effect of age, you need to take into account the differences in educational levels between younger and older people. You need to mobilize data and other studies on Armenia's socio-cultural context.

We added “This may be due to a lower education levels in older adults, although a majority (70.5%) of our sample had at least a university or vocational degree.” (lines 273-277)

7. Finally, The authors need to add limitations regarding the type of survey (phone survey). In addition, I suggest that the authors add a section ‘Policy implication of findings’

We would like to explain why the type of the survey being a phone survey might not be considered a limitation for this study. Out of 3,727 people who underwent blood sampling for the antibody testing 3,483 (93.5%) participated in the phone survey. The study participants gave written informed consent that after the blood sampling they will be contacted by the research team for a phone survey. However, it might be that those who agreed to give a blood sample for the bigger study might be more concerned and aware of their health and had a higher degree of health literacy and knowledge compared to those who refused to participate. To acknowledge this, we added in the limitations that “Also, those more concerned about their health and/or more aware of infectious diseases might agree to participate in the blood sampling for COVID-19 antibody testing introducing some degree of self-selection bias”. (lines 299-301) We also added a section on the Policy implications. (lines 311-328)

Reviewer #2: 

Employing a comprehensive nationwide phone survey, the study endeavours to measure the extent of health literacy and knowledge concerning infectious diseases across Armenia, while also examining the impact of socio-economic factors on these variables.

The article is generally well-written, and it includes a thorough review of the literature, presents a sound methodology, and discusses the findings in a comprehensive manner.

Some minor revisions can be addressed to enhance the credibility and interpretability of the research findings; details are reported below.

Thank you!

1. I believe that an evidence-based definition of “working from home” and a consistent use of the related terminology for this exposure.

We would like to explain why we have used “stay at home from work/school” in some infectious disease-related literacy questions in our instrument. The questions were taken from the WHO survey tool designed to guide behavioural insights studies related to COVID-19 (https://iris.who.int/bitstream/handle/10665/333549/WHO-EURO-2020-696-40431-54222-eng.pdf?sequence=1&isAllowed=y). We did not change the wording as the similar wording was used by the government (Ministry of Health) to communicate the COVID-19 related preventive recommendations. We used the same terminology/wording, so the questions are understandable to our survey participants representing general population of Armenia. 

2. The study design should be understood from the title and the abstract; it should bel also mentioned in the methods section.

We added the study design in the title: “Socio-demographic determinants of infectious disease-related health literacy and knowledge in Armenia: Results from a nationwide survey”. (Lines 1-3)

3. Even though the language adopted in the article is correct, I suggest refining the style to achieve better clarity and precision. Be careful in ensuring that the terminology used is the most accurate and specific to avoid ambiguity.

We edited throughout the manuscript to better refine the style and clarity.

4. The methodology is well-structured, employing a nationwide phone survey to collect data. However, the study could benefit from a more detailed description of the survey questions and the rationale behind the selection of these specific questions. Clarification on how the questions were tailored to assess health literacy and knowledge specifically related to infectious diseases would strengthen the paper.

We added in the Data collection and study tool section explanation. “The infections disease-related health literacy questions were adapted to from the WHO survey tool designed to guide behavioural insights studies related to COVID-19 to cover general infectious disease-related health literacy.” (Lines 151-159)

5. It's mentioned that the scale used to assess infectious disease-related health literacy was not previously validated, which could affect the reliability of the results. If possible, use a validate scale or, in future studies, consider validating the scale used in this paper.

Thank you for this note. Given that there was (is) no validated tool available in Armenian language, the infections disease-related health literacy questions were taken from the WHO survey tool designed to guide behavioral insights studies related to COVID-19. However, we adapted the scale to cover not only COVID-19 but infectious disease literacy in general and translated to Armenian.

6. While the paper acknowledges several limitations, including the potential bias introduced by phone surveys and the lack of validation for the health literacy scale, a more thorough exploration of these limitations and their implications for the study's findings would be beneficial. Additionally, discussing potential strategies for overcoming these limitations in future research could provide a more comprehensive picture.

We added on the recommendations for future similar studies "Future studies may consider using validated scales and designing comprehensive questionnaires specifically focusing on the infectious disease-related health literacy and knowledge" (Lines 306-307)

7. A detailed participant flowchart or description of participant recruitment, inclusion, and exclusion criteria would enhance transparency in reporting.

We added a reference to a study which explains the participant recruitment, inclusion, and exclusion criteria in more details (lines 125-126).

8. The discussion effectively highlights the importance of the study findings in the context of Armenia and beyond. However, the paper would benefit from a broader consideration of the implications for public health policy and education strategies. Specifically, recommendations for how policymakers and educators can use these findings to improve health literacy and preparedness for future infectious disease outbreaks would be useful.

We added a section on the Policy implications. (lines 311-328)

---

## [Decision Letter · Decision Letter 1]

1 Jul 2024

Socio-demographic determinants of infectious disease-related health literacy and knowledge in Armenia

PONE-D-23-40060R1

Dear Dr. Sargsyan,

We’re pleased to inform you that your manuscript has been judged scientifically suitable for publication and will be formally accepted for publication once it meets all outstanding technical requirements.

Kind regards,

Laura Brunelli, MD, PhD

Academic Editor

PLOS ONE

Additional Editor Comments (optional):

Reviewers' comments:

Reviewer's Responses to Questions

**Comments to the Author**

1. If the authors have adequately addressed your comments raised in a previous round of review and you feel that this manuscript is now acceptable for publication, you may indicate that here to bypass the “Comments to the Author” section, enter your conflict of interest statement in the “Confidential to Editor” section, and submit your "Accept" recommendation.

Reviewer #1: All comments have been addressed

Reviewer #2: All comments have been addressed

2. Is the manuscript technically sound, and do the data support the conclusions?

Reviewer #1: Yes

Reviewer #2: Yes

3. Has the statistical analysis been performed appropriately and rigorously? 

Reviewer #1: Yes

Reviewer #2: Yes

4. Have the authors made all data underlying the findings in their manuscript fully available?

Reviewer #1: Yes

Reviewer #2: Yes

5. Is the manuscript presented in an intelligible fashion and written in standard English?

Reviewer #1: Yes

Reviewer #2: Yes

6. Review Comments to the Author

Reviewer #1: For p-values in tables, replace p=0.0000 with p<0.0001. The p-values are never zero, but rather close to 0. This is my only comment for the authors. Before the article is published, they must correct this.

Reviewer #2: Dear Authors,

I have carefully reviewed your responses to my comments and the revised version of your manuscript. I am pleased to inform you that your thorough revisions have adequately addressed all my concerns. The manuscript is now of a high standard and is suitable for publication. I commend you for your diligent work and thoughtful revisions.

Best regards

7. PLOS authors have the option to publish the peer review history of their article (what does this mean?). If published, this will include your full peer review and any attached files.

Reviewer #1: No

Reviewer #2: No

---

## [Editor Report · Acceptance letter]

10 Jul 2024

PONE-D-23-40060R1 

PLOS ONE

Dear Dr. Sargsyan, 

I'm pleased to inform you that your manuscript has been deemed suitable for publication in PLOS ONE. Congratulations! Your manuscript is now being handed over to our production team.

Kind regards, 

on behalf of

Dr. Laura Brunelli 

Academic Editor

PLOS ONE